# Acetic-Acid Plasma-Polymerization on Polymeric Substrates for Biomedical Application

**DOI:** 10.3390/nano9070941

**Published:** 2019-06-28

**Authors:** Shu-Chuan Liao, Ko-Shao Chen, Jui-Lung Chien, Su-Chen Chen, Win-Li Lin

**Affiliations:** 1Institute of Biomedical Engineering, National Taiwan University, Taipei 106, Taiwan; 2Bachelor Program for Design and Materials for Medical Equipment and Devices, Da Yeh University, Changhua 515, Taiwan; 3Department of Materials Engineering, Tatung University, Taipei 104, Taiwan; 4Department of Raw Materials and Yarns, Taiwan Textile Research Institute, New Taipei City 236, Taiwan

**Keywords:** acetic acid, plasma polymerization, hydrophilic, surface modification

## Abstract

Cold plasma is an emerging technology offering many potential applications for regenerative medicine or tissue engineering. This study focused on the characterization of the carboxylic acid functional groups deposited on polymeric substrates using a plasma polymerization process with an acetic acid precursor. The acetic acid precursor contains oxygen and hydrocarbon that, when introduced to a plasma state, forms the polylactide-like film on the substrates. In this study, polymeric substrates were modified by depositing acetic acid plasma film on the surface to improve hydrophilic quality and biocompatibility. The experimental results that of electron spectroscopy for chemical analysis (ESCA) to show for acetic acid film, three peaks corresponding to the C–C group (285.0 eV), C–O group (286.6 eV), and C=O group (288.7 eV) were observed. The resulting of those indicated that appropriate acetic acid plasma treatment could increase the polar components on the surface of substrates to improve the hydrophilicity. In addition, in vitro cell culture studies showed that the embryonic stem (ES) cell adhesion on the acetic acid plasma-treated polymeric substrates is better than the untreated. Such acetic acid film performance makes it become a promising candidate as the surface coating layer on polymeric substrates for biomedical application.

## 1. Introduction

Polymers have generated considerable interest as biomaterials in the field of tissue engineering, such as for tissue replacement, tissue reinforcement and organ transplantation [1]. Poly (dl-lactide-co-glycolide)s (PLGAs) have been widely used in the preparation of scaffolds for tissue engineering applications because of their good biocompatibility [1,2,3]. Although these materials have been approved by the Food and Drug Administration (FDA), most of them are too flexible or too weak to meet the mechanical strength and consumption requirements when preparing the scaffolds [4,5,6]. Polymers such as polypropylene (PP) nonwoven and polyethylene terephthalate (PET) film are suitable scaffolds for applications in tissue engineering, owing to their mechanical strength and easy manipulation into desired shapes. However, foams, from such synthetic polymers, are hydrophobic in nature and lack cell recognition signals. Surface modification of polymers by plasma treatment is industrially attractive, as the technique is simple, easy to implement, reliable, non-polluting, and cost effective [7]. PLGA is a bioresorbable material due to good biocompatibility, controllable biodegradability, and relatively good processability. However, it has poor hydrophilicity, and there are no natural cell recognition sites on the surface of poly(lactide-co- glycolide). Gas plasma treatment is extensively used for chemical modification of polylactide (PLLA) or poly (lactide-co-glycolide) (PLGA).Plasma processes have been used to increase the hydrophilicity of polylactide (PLA) and to improve its cell adhesion. This investigation aims to quickly prepare the PLGA-like surface and expect good biocompatibility using acetic acid plasma technology [8,9].

In order to enhance other applications of the materials, and without there being any deterioration in the bulk properties, surface modification plays a very important role in various fields [8,9,10,11,12,13]. There are several methods that have been considered and developed for altering the interactions of materials with their environments, such as adsorption, oxidation by strong acids, ozone treatment, plasma (glow discharge), corona discharge, photo activation (UV), ion, electron beam, and so on [14,15,16,17]. Among these methods, plasma surface modification processes account for most of the commercial uses of plasma technology because they are fast, efficient methods for increasing the adhesion, wettability properties and other surface characteristics of a variety of materials [7,18,19,20]. A major advantage of plasma surface modification, as compared with most other treatments, is that it is free of harmful sub-products from the operation process and does not destroy the bulk structure of materials. Synthetic polymers therefore often require selective modification to introduce specific functional groups to the surface for improving the biocompatibility [21]. A desired monomer may be polymerized onto the surface of a plasma-activated polymer, resulting in the formation of a grafted brush layer on the surface [16,22]. The surfaces may then provide polar components for increasing biocompatibility. This method is highly surface-selective, where the modification is confined to a depth of a few nanometers without modification of the bulk properties [23,24,25]. Several researchers have studied polymer grafted with acrylic acid using oxygen plasma in order to immobilize insulin and heparin on the surface (such as PDMS, PET, PU, PLGA etc) [24,26,27]. In particular, the surface structure and dynamics of the plasma-grafted polymer have not been properly addressed in the context of polymer surface-cell interactions. Therefore, it is extremely interesting to monitor eventual alterations in surface morphology, as they are likely to considerably affect cell growth. In order to maintain a surface with a high level of hydrophilicity and functionality at the surface, subsequent modification by grafting of organic acid onto a plasma treated polymer was carried out [28,29,30,31]. Plasma polymerization has consequently proven highly successful as a means of developing functional interfaces for cell culture [27,32,33].

In this study, a plasma-polymerization film of acetic acid was deposited on the surface of polymers, such as polypropylene (PP) nonwoven, polyethylene terephthalate (PET) and Poly (dl-lactide-co-glycolide) by plasma polymerization, to improve the hydrophilic quality and biocompatibility. The specimens subjected to various treatments were characterized by water contact angle, Fourier Transformation Infrared (FTIR), Electron Spectroscopy for Chemical Analysis (ESCA) and the cell viability against embryonic stem (ES) cell adhesion

## 2. Materials and Methods

### 2.1. Pretreatment of Materials

Polypropylene (PP) nonwoven and polyethylene terephthalate (PET) were offered from Textile Research Institute, New Taipei city, Taiwan and cut into a suitable size of about 2 × 2 cm^2^. The substrates were ultrasonically-cleaned with deionized water, then ethanol, for 10 min in each solution and dried in a desiccator to remove contaminants and organic matters on the surface. Poly (dl-lactide-co-glycolide) (PLGA, 75:25) was obtained from Bio Invigor corporation, Taipei, Taiwan. PLGA solution was prepared by dissolving PLGA polymer in chloroform. NH_4_HCO_3_ (Mw = 79.055 g/mol, Sigma, St. Louis, MO, USA) /NaCl (Mw = 58.44, g/mol, Sigma, St. Louis, MO, USA)sieved salt particulates (particle size 125–180 μm) were added to PLGA solution. The mixture of polymer/salts/solvent was cast into a glass slide (2.54 × 7.62 cm^2^) as a sheet mold. The sheet was de-molded after being air-dried for 1.5 h and then vacuum dried for 1 h. The sheet was first immersed in 90 °C hot water to leach out the NH_4_HCO_3_ particles and was subsequently immersed in 60 °C hot water to leach out the NaCl particles. The sheet was freeze-dried and stored in a desiccator.

### 2.2. Cold Plasma Treatment

The plasma polymerization equipment had a bell-jar reaction chamber and 13.56 MHz radio frequency generators (model PD-2S manufactured by SAMCO Co., Kyoto, Japan) [16]. A vacuum pump was employed to provide a low-pressure environment. The cleaned polymeric substrates’ specimens were placed on the lower electrode of the reaction chamber before being evacuated. The reaction chamber was evacuated to less than 30 mTorr. The argon gas was introduced into the reaction chamber and maintained at a constant pressure by adjusting the micro-throttle valve in the argon-inlet tube. After the pressure was stabilized, the specimen surface was cleaned at an input power of 20 W using argon gas of 100 mTorr for 3 min. In order to obtain a carboxylic acid group on the polymeric substrates surface, the surface must be coated with acetic acid precursor by a cold plasma process. The monomers acetic acid was purchased from Fluka Chemical Co., New York, NY, USA, ((CH_3_COOH), 95%, Mv = 60 g/mol). The reactor chamber pressure was controlled by gas input and vacuum pump throttle valves through which acetic acid monomers were introduced into the plasma chamber at a pressure of 100, 150 and 200 mTorr, respectively. As the gas input was controlled by monomer evaporation, exact calculation of the flow rate was not possible. The processing power was at either 10, 30 or 50 W for treatment times of 10 min, respectively. Figure 1 shows the schematic illustration of the preparation of the polymeric film surface modification with acetic acid plasma, and then cell culture.

### 2.3. Characteristics Analysis

#### 2.3.1. Wettability (Surface Hydrophobicity/Hydrophilicity) Test

Surface wettability of the untreated and the acetic acid plasma deposited films were measured by the sessile drop (0.9 μL) method with distilled water by a syringe and observed by CCD at room temperature (25.0 °C) (Goni-meter type G-1 made by ERMA Optical Works Co. LTD, Tokyo, Japan,). The drop image was recorded by a video camera. The measured water contact angles (WCA) value was the average of five measurements.

#### 2.3.2. Chemical Composition

The functional groups in polymerized films were analyzed by Fourier transformation infrared (FT-IR). The characteristic absorption peaks were detected using a JASCO FT/IR-300E (Tokyo, Japan) spectrometer. The chemical composition and bonding of the coating was obtained by electron spectroscopy for chemical analysis (ESCA) using a ULVAC-PHI PHI 5000 Versaprobe II (Kanagawa-ken, Japan). All of the binding energy of photoelectrons at the emission angle were referenced to a CHx peak at the maximum resolved C1s peak at 285.0 eV.

#### 2.3.3. Cell Culture

The biocompatibility of the acetic acid plasma film on the surface was assessed by in-vitro cell culture, wherein the substrates (0.5 × 0.5 cm^2^) were washed using PBS solution and placed into a 24 well plate. J1 mouse embryonic stem cells (ATCC^®^ SCRC-1010) were obtained from the American Type Culture Collection (Union Biomed INC, Taiwan). The embryonic stem (ES) cells were cultured in medium Dulbecco’s modified Eagle medium (DMEM, Gibco, Thermo Fisher Scientific, Waltham, MA, USA) supplemented with 10% fetal bovine serum (FBS, Gibco, USA), 100 μg/mL streptomycin, 300 μg/mL β-mercaptomethanol and leukocyte inhibitory factor (LIF, Gibco, USA). ES cells were dissociated into a single-cell suspension with 0.25% trypsin, maintained in a gas-jacket incubator with 5% CO_2_ at 37 °C. Embryonic stem cells, a recommended line of cells, were used for cytotoxicity and cell-substratum interactions with biomaterials studies. Identification of the effects of acetic acid plasma treatments on early cell adhesion and cell viability was carried out using the MTT assay. After incubation, 1 × 10^6^ cells mL^−1^ in 100 µL of DMEM supplemented with 10% FBS were seeded to the wells containing different plasma treated substrates, and were cultured for 10–14 days.

The cell culturing time was set at 1, 3, 7 and 10 days at 37 °C to enable cell adhesion and direct cell viability assays, respectively. Subsequently, the cells were washed twice with 0.1 M phosphate-buffered saline (PBS) pH 7.4 at 37 °C and incubated with 100 µL of medium. MTT assay mixture (Sigma, St. Louis, MO, USA) was added to each well, and the culture dish was wrapped in an aluminum foil and incubated for 4 h at 37 °C. After 4 h, the culture medium in each well was substituted with 100 µL of Dimethyl Sulphoxide (DMSO, St. Louis, Missouri, USA) and 25 µL of buffered Glycine/Sorensen solution to dissolve formazan crystals. The optical density (OD) at a wavelength of 570 nm was measured using an enzyme-linked immunosorbent assay (ELISA) reader. The cell viability assays were presented as mean ± standard deviation (*n* = 3).

### 2.4. Cell Morphology Analysis

For cell adhesion experiments, ES cells were gently washed with PBS after 7 days of growth. Cells were fixed in 4% formaldehyde for 20 min at room temperature and permeabilized with 0.1% Triton X-100 in PBS for 15 min at 20 °C. After staining, cell morphology and distribution were observed and photographed by a fluorescence microscope (Nikon TE300, Tokyo, Japan).

In order to carry out cell morphology analysis approximately 1 × 10^6^ ES cells mL^−1^ of medium DEME supplemented with 10% FBS were seeded in each well containing the acetic acid plasma treatment and cultured at 37 °C, 5% CO_2_. At the harvest point of 7 days, the medium was removed and washed with PBS. The substrates were fixed with 2.5% glutaraldehyde for 1 h at room temperature. Then, the substrates were rinsed with PBS again, and dehydrated sequentially in 30%, 50%, 75%, and 100% ethanol for 10 min. Gold was coated onto the specimens to allow the cells to be conductive using a sputter coater, and the substrates were observed using SEM (JEQL JSM-6300, Tokyo, Japan).

## 3. Results and Discussion

### 3.1. Wettability of Surface-Modified PET and PP Nonwoven

Several researchers have found that hydrophilicity is an advantage for bioactive materials. It can improve cell attachment, spreading, and proliferation [29,30]. The water contact angle results of various polymeric substrates with different treatments are shown in Table 1. From the results, it can be seen that the surface of substrates treated by acetic acid plasma was more hydrophilic than when it was untreated; accordingly, the water contact angle was reduced after acetic acid plasma treatment. The intrinsically hydrophobic PP nonwoven, PET and PLGA can be modified into a hydrophilic surface following acetic acid plasma treatment.

Based on the stability of the functionalized surface, decay degree of hydrophilicity of acetic acid plasma-deposited films is shown in Figure 2. The water contact angle of untreated PET film was 67.5° ± 4.5° accordingly, the water contact angle of PET after acetic acid plasma treatment (AAP) and Ar plasma treatment was reduced (AAP: 67.5° ± 4.5° to 8.2° ± 2.5°, Ar: 67.5° ± 4.5° to 32° ± 3.4°, plasma modified condition: AAP 50 W, 10 min, 200 mTorr). After again effect, the hydrophilicity of the PET after Ar plasma treatment decreased from the increasing aging time. It can be inferred that the aromatic compound benzene in the structure of PET is stable. As a result, a more hydrophilic and stable surface after acetic acid plasma-deposited films was observed on the PET surface. The intrinsically hydrophobic polypropylene nonwoven could be modified into a hydrophilic surface after acetic acid plasma treatment. The wettability ratio of nonwoven was increased approximately to 420 wt% after acetic acid plasma polymerization showed in Table 2. It could be observed that the acetic acid plasma treatment could effectively increase hydrophilic property of the surface.

### 3.2. FTIR Characterization of Acetic Acid Membrane Structure

Figure 3 shows the FTIR spectrum of PLGA treated by acetic acid plasma treatment under different conditions. It could be observed that several adsorption peaks of the acetic acid film appeared or were enhanced; for example, C=C at 1600 cm^−1^, C–O at 1125 cm^−1^, C–H at 1424 cm^−1^, –OH at 3050~3500 cm^−1^ and C=O at 1700~1720 cm^−1^ were reduced. As the plasma power increases, the peaks pertaining to the acetic acid film functional groups become stronger.

### 3.3. Chemical Analysis of Acetic Acid Film

ESCA analysis is used to characterize the chemical bonding and functional groups of the surface film. Turning now to the experimental evidence on chemical analysis of acetic acid film. In this study, a Si wafer was used as the substrate for ESCA analysis. Table 3 and Table 4 showed the atomic composition by ESCA of the various polymerizations of the acetic acid plasma-deposited films on the Si-wafer surface. Oxygen/carbon ([O]/[C]) ratio, obtained by integrating the O 1’s and C 1’s peak atomic mole fraction % is shown in Table 3. A considerable increment of the oxygen atomic mole fraction % with the proceeding of decreasing power of plasma treatment was observed and [O]/[C] ratio while that was large. In the ESCA analysis the carbon (C1s) spectra of substrates after acetic acid plasma-deposited films in different wattages are presented in Figure 4a–c. For acetic acid film, three peaks corresponding to the C–C group (285.0 eV), C–O group (286.6 eV), and C=O group (288.7 eV) were observed. Relative intensity (% area) of various C–C, C–O and C=O functional groups treated by acetic acid plasma treatment with different plasma conditions was showed in Table 4. An increase in the C=O groups was observed while the wattage of plasma treatment decreased. The increase in the functional groups containing oxygen has been reflected on the change in the surface properties of the substrate. The improvement to wettability may occur to plasma deposition PLGA-like film. This indicated that appropriate acetic acid plasma treatment could increase the polar components on the surface of substrates to improve the hydrophilicity and biocompatibility.

### 3.4. Cell Morphology Observations

In the cell culture periods of 3 and 7 days, morphology of ES cell attachment to PP nonwoven and PLGA after acetic acid plasma treated was observed by microscope. The cell structure and growth were further evaluated by fluorescence microscopy image (in green color). Figure 5 shows the ES cell after 3 and 7 days of growth on PP nonwoven and PLGA fabricated with acetic acid plasma, together with the ES cell on untreated for comparison. It can be seen that control PP nonwoven and PLGA served as a poor substrate for cell growth, and the attached area of the cells is limited. The cultured cells began to attach to and proliferate on the surface treated by acetic acid plasma treatment within 3–7 days. The cells grown in direct contact with the surface and spread out uniformly can be clearly observed. On the basis of these results in Figure 5, acetic acid plasma-deposited films on PP nonwoven and PLGA show good biocompatibility and all are more so than the untreated. The cells on the surface of the acetic acid film appear distinctly to be more extensive in coverage which suggests that acetic acid film provides a permissive surface for cells adhesion and grown.

The acetic acid plasma treated surfaces were imaged by SEM to observe ES cell confluence as shown in Figure 6 after 7 days in culture appeared as layers covering the PLGA surface. From Figure 6, it can be seen that cells on PLGA were round and seemed to be adhering, showing that acetic acid plasma treatment to improves support adhesion and proper cell division and growth.

### 3.5. In Vitro Cytocompatibility Assay

As seen in Figure 7a,b, MTT assay revealed that ES cell adhesion and growth on PP nonwoven and PLGA with an acetic acid plasma-deposited are superior to untreated PP nonwoven and PLGA. As per the recorded cell density, it is clear that the cell proliferated on the acetic acid plasma-deposited on samples at higher levels than the uncoated samples. Significant difference can be observed between the cell attachment on acetic acid films coating and control from the 1 day of the culture time, confirming that the cells could augment well on acetic acid films coating layer. Acetic acid plasma deposition can increase hydrophilicity on the surface, which specifies the higher bioactivity of the coated surface and it can increase the cell compatibility of the materials. That is the reason why cells adhere to the acetic acid plasma deposited on PP nonwoven and PLGA better than untreated PP nonwoven and PLGA. In this study, the polymeric substrates modified by acetic acid plasma deposition at 30 W and 200 mTorr for 10 min showed the best cytocompatibility assay.

## 4. Conclusions

In this study, the wettability results indicate the substrate following acetic acid plasma deposition will improve the hydrophilic quality of substrate efficiency. Hydrophilicity of the Ar plasma treated film decreased from the increasing aging. On the other hand, the substrates following acetic acid plasma treatment still retained their hydrophilic nature. FTIR spectra demonstrated that acetic acid film was successfully immobilized onto the surface of scaffolds. ESCA spectra showed that appropriate acetic acid plasma treatment could increase the polar components on the surface of substrates to improve the hydrophilicity. In addition, cell adhesion and growth of acetic acid plasma-treated scaffolds (PLGA and PP nonwoven) were more active than on the untreated scaffolds. On the untreated scaffolds, cells did not adhere to the surface, while cells were dispersed onto the acetic acid plasma-treated scaffold’s surface. The MTT assay results showed that ES cell adhesion and growth on the acetic acid plasma deposited surface of scaffolds is superior to the untreated scaffolds. Therefore, acetic acid plasma treatment gives us the unique capability of modifying the prosthetic biomaterials of various constructs with the eventual transplantation of mammalian cells to be used in tissue engineering or as biomedical applications.

## Figures and Tables

**Figure 1 nanomaterials-09-00941-f001:**
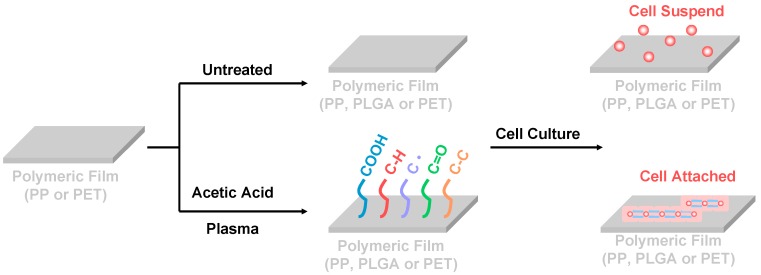
The schematic illustration of the preparation of polymeric film surface modification with acetic acid plasma and then cell culture.

**Figure 2 nanomaterials-09-00941-f002:**
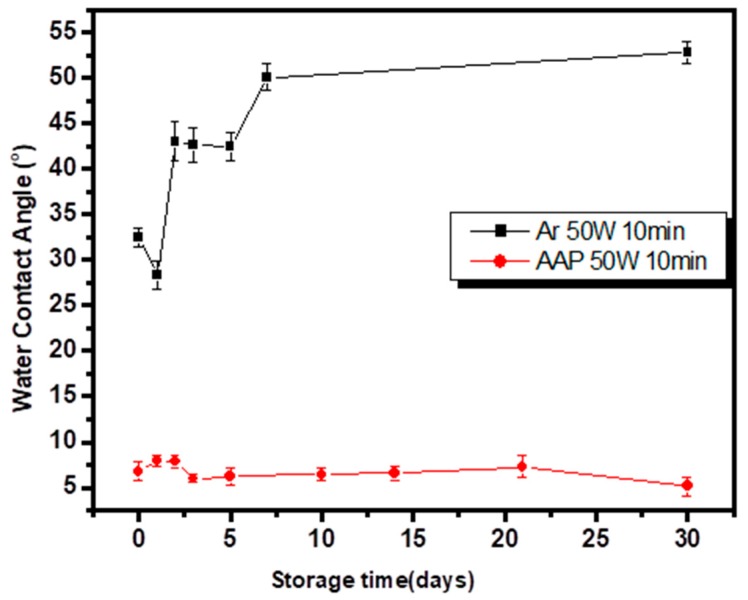
Aging effect of the plasma-treated polyethylene terephthalate (PET) surface hydrophilicity.

**Figure 3 nanomaterials-09-00941-f003:**
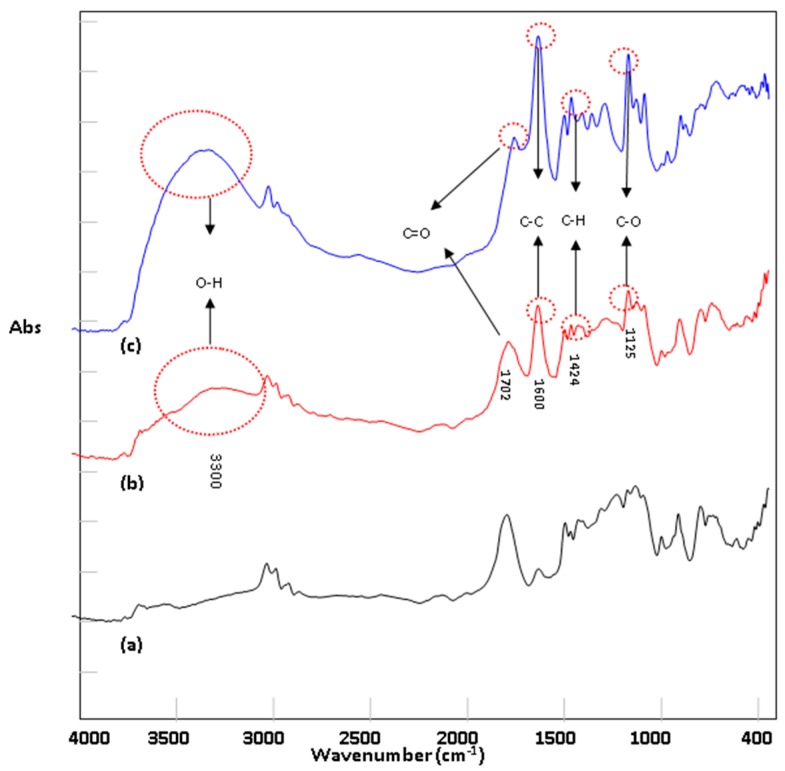
FTIR spectra of acetic acid plasma modified poly (dl-lactide-co-glycolide)s (PLGAs) (**a**) untreated; (**b**) treated with AAP: 30 W, 10 min; (**c**) treated with AAP: 50 W, 10 min.

**Figure 4 nanomaterials-09-00941-f004:**
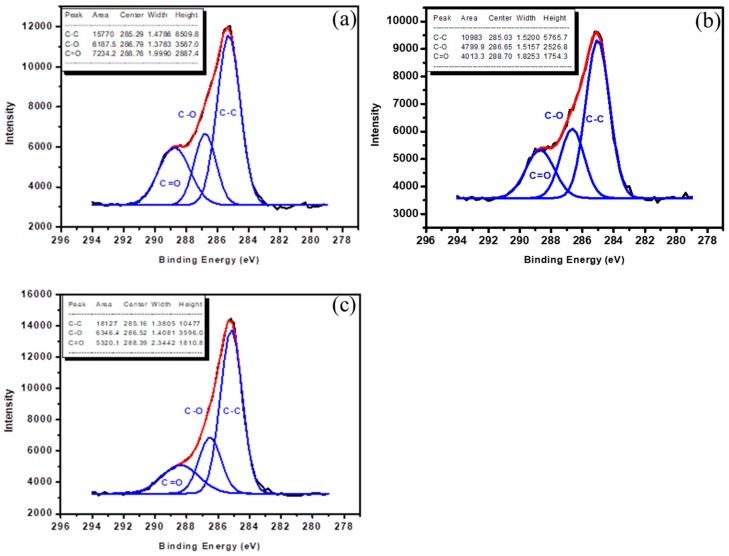
Electron spectroscopy for chemical analysis (ESCA) C1s spectra of (**a**) acetic acid film (10 W, 10 min), (**b**) acetic acid film (30 W, 10 min), (**c**) acetic acid film (50 W, 10 min).

**Figure 5 nanomaterials-09-00941-f005:**
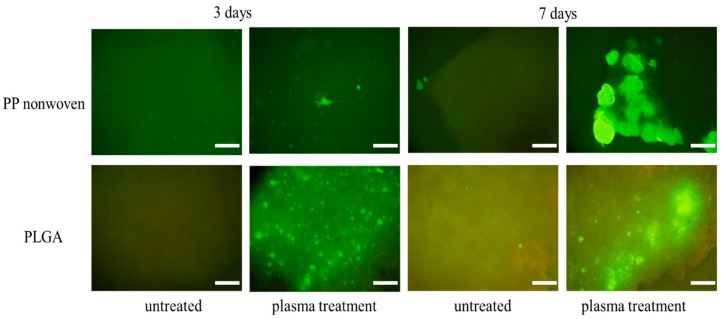
After acetic acid plasma treatment, embryonic stem (ES) cells on the PP nonwoven and PLGA was observed using fluorescence microscopy. Scale bars represent 50 μm.

**Figure 6 nanomaterials-09-00941-f006:**
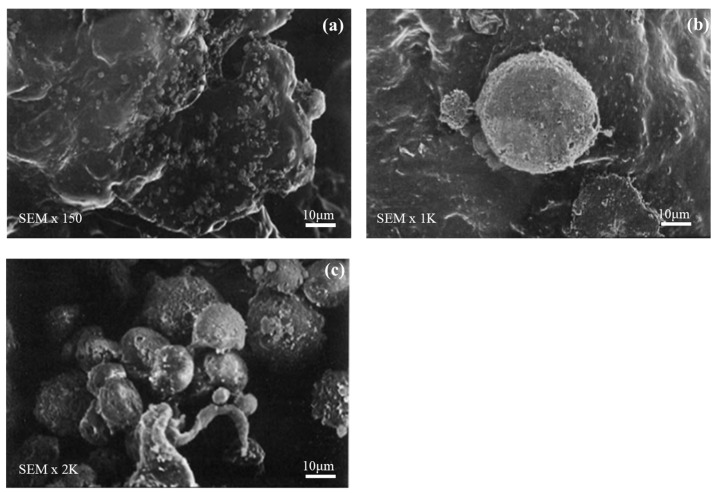
SEM micrographs: (**a**) ES cells of PLGA (150); (**b**) ES cells of PLGA (1 K); and (**c**) ES cells of PLGA (2 K).

**Figure 7 nanomaterials-09-00941-f007:**
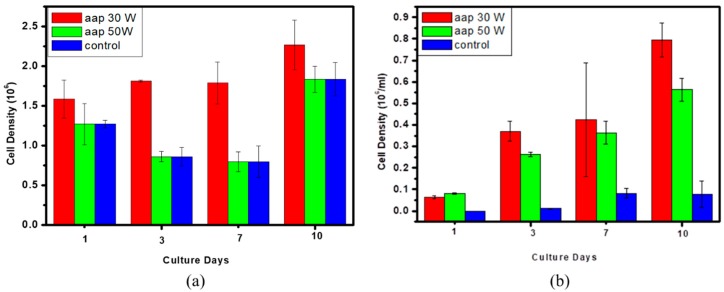
Cell density of ES cells for in vitro cytocompatibility assay for days 1, 3, 7 and 10 for the PP nonwoven and PLGA without and with the acetic acid films deposited at different plasma powers. (**a**) PP nonwoven (**b**) PLGA.

**Table 1 nanomaterials-09-00941-t001:** Wettability of acetic acid plasma-treated substrates.

Substrates	Control	Acetic Acid Plasma Treated
PP	85.5 ± 6.9	12.1 ± 1.5
PET	67.5 ± 4.5	9.6 ± 2.3
PLGA	58.0 ± 4.5	28.8 ± 4.1

Plasma Modified Condition: aap 30 W, 10 min, 200 mTorr.

**Table 2 nanomaterials-09-00941-t002:** The wettability of polypropylene (PP) nonwoven after treatment (pre-treatment is hydrophobic) (by absorption H_2_O).

	Dried	Immerse in Water	Absorb Ratio (%)
**Untreated Weight (mg)**	9.2	9.2	0
**After Plasma treated 100 mTorr Weight (mg)**	9.5	47.5	400
**After Plasma treated 150 mTorr Weight (mg)**	10.2	50.9	46.9
**After Plasma treated 200 mTorr Weight (mg)**	9.1	46.9	420

Plasma Modification Condition: aap 30 W 10 min. The extent of water adsorption (%) = [(W2 − W1)/W1] × 100%. Where W1 and W2 denote the weight of the nonwoven and nonwoven immersed in water, respectively.

**Table 3 nanomaterials-09-00941-t003:** The atomic mole fraction of O, C and O/C on Si wafer after acetic acid plasma is deposited under different conditions.

	Atomic Mole Fraction %		
	C	O	O/C
10 W, 10 min	54.35	45.65	0.84
30 W, 10 min	68.85	31.15	0.45
50 W,10 min	76.45	23.55	0.31

**Table 4 nanomaterials-09-00941-t004:** Relative intensity (% area) of various functional groups (chemical bonds) measured by the C1s peaks from the Si wafer surfaces treated by acetic acid plasma with different conditions.

Function Groups	10 W, 10 min	30 W, 10 min	50 W, 10 min
C–C	54.02%	55.48%	60.84%
C–O	21.20%	24.25%	21.30%
C=O	24.78%	20.27%	17.86%

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
