# Peer review of "Acetic-Acid Plasma-Polymerization on Polymeric Substrates for Biomedical Application"

_nanomaterials, 2019, doi:10.3390/nano9070941_

Reviewer 1 Report

The majority of issues mentioned by the reviewers was considered by the authors in the revised version.

However, for the following reason I  can not recommend this manuscript for publication:

Contrary to the statement in the author reply (“The source of the cells have been added in section 2.3.3”) the only information regarding cell type and source is “embryonic stem cells, a recommended line of cells”. No reference is cited for this statement. No details are given.      

Author Response

Dear Editors and Reviewers:

Thank you for your letter and for the reviewers’ comments concerning our manuscript entitled “Acetic-Acid Plasma-Polymerization on polymeric substrates for biomedical application” (nanomaterials-524969). Those comments are all valuable and very helpful for revising and improving our paper, as well as the important guiding significance to our researches. We have studied comments carefully and have made correction that we hope meet with approval. A revised manuscript with the correction sections red marked was attached as the supplemental material and for easy check/editing purpose. Should you have any questions, please contact us without hesitate. The main corrections in the paper and the responds to the reviewer’s comments are as flowing:

Responds to the reviewer’s comments:

Reviewer #1:

1. Response to comment: The majority of issues mentioned by the reviewers was considered by the authors in the revised version. However, for the following reason I cannot recommend this manuscript for publication: Contrary to the statement in the author reply (“The source of the cells have been added in section 2.3.3”) the only information regarding cell type and source is “embryonic stem cells, a recommended line of cells”. No reference is cited for this statement. No details are given.

The authors’ Answer: We are very sorry for our negligence of in our manuscript. We have revised our about the source of the cell the revised part in manuscript is following: The source of the cells have been added in section 2.3.3, J1 mouse embryonic stem cells (ATCC® SCRC-1010) were obtained from the American Type Culture Collection. We have add more references[26-28,31] in the manuscript.

All the lines and pages indicated above are in the revised manuscript. Thank you and all the reviewers for the kind advice.

Sincerely yours,

Reviewer 2 Report

The study is well designed and the paper is well structured. State of the art is appropiated. Experimental methods are detailed. Discussions are supported by results. The paper can be proposed for publication.

Author Response

Dear Editors and Reviewers:

Thank you for your letter and for the reviewers’ comments concerning our manuscript entitled “Acetic-Acid Plasma-Polymerization on polymeric substrates for biomedical application” (nanomaterials-524969). Those comments are all valuable and very helpful for revising and improving our paper, as well as the important guiding significance to our researches. We have studied comments carefully and have made correction that we hope meet with approval. A revised manuscript with the correction sections red marked was attached as the supplemental material and for easy check/editing purpose. Should you have any questions, please contact us without hesitate.

The study is well designed and the paper is well structured. State of the art is appropriated. Experimental methods are detailed. Discussions are supported by results. The paper can be proposed for publication.

The authors’ Answer: Thank you for your mentions. I am very grateful to your comments for the manuscript.

All the lines and pages indicated above are in the revised manuscript. Thank you and all the reviewers for the kind advice.

Sincerely yours,

Round  2

Reviewer 1 Report

The majority of issues mentioned by the reviewers was considered by the authors in the first version. The source of cells was added in the second revision. 

This manuscript is a resubmission of an earlier submission. The following is a list of the peer review reports and author responses from that submission.

Round  1

Reviewer 1 Report

The study "Study on Improvement of Stem cell Adhesion on Acetic-Acid Plasma-Polymerization treated Film" shows the effect of the plasma treatment with acetic acid onto substrates of PET, PP and PLGA. 

Despite the soundness of the research migh be interesting, the paper is completely unstructured and lacks of information in some sections. Apart form it, english must be revised since the discussion is hard to follow. Some recommendations to improve the paper for a deeper revision:

1) Explain in all cases the effect of plasma on the three substrates. Some times it seems that is in only applied to PP and PET (see aim), other to the three substrates (see Fig 1), other to PLGA (see "Materials"!), others to just PLGA and PP (section 3.4). It does not help for comparison.

2) Explain all experimental procedures in detail. What about FTIR, ESCA or wettability? What about

3) What is the interest of the deconvolution studies on FTIR bands? Are they correlated to a better or worse cell adhesion?

4) Show the main conclusions as a synthesis of main results, and align it in the abstract. In this sense, try to change qualitative terms such as "better" or "worse", and changed them by quantitative results, whwerever possible.

Reviewer 2 Report

The manuscript addesses an important topic.

However, the manuscript does not meet basic requirements of a scientific publication.

I give only some examples:

Type and source of materials are not specified (PET, PLGA).

The source of the cells is not specified.

Type and operation mode of analytical instruments are not specified (contact angle measurements, FTIR, ESCA).

Abbreviations are not introduced (PET, FTIR, MTT).

Scale bars are missing in micrographs (Fig. 8).

non-SI units are used (Torr)